# Reduced Carbohydrate Diet Influence on Postprandial Glycemia—Results of a Short, CGM-Based, Interventional Study in Adolescents with Type 1 Diabetes

**DOI:** 10.3390/nu14214689

**Published:** 2022-11-05

**Authors:** Agnieszka Lejk, Jędrzej Chrzanowski, Adrianna Cieślak, Wojciech Fendler, Małgorzata Myśliwiec

**Affiliations:** 1Department of Pediatrics, Diabetology and Endocrinology, Medical University of Gdansk, 80-211 Gdansk, Poland; 2Department of Biostatistics and Translational Medicine, Medical University of Lodz, 92-215 Lodz, Poland

**Keywords:** pediatric diabetes, carbohydrates, individualized nutritional guideline

## Abstract

Therapy for type 1 diabetes (T1DM) focuses on maintaining optimal blood glucose levels, achieved with intensive insulin treatment, proper nutrition, and physical activity. The aim of this study was to investigate postprandial glycemic changes under low (30%) and standard (50%) carbohydrate diets in adolescents with T1DM. A single-center cross-over nutritional study was conducted, during which 26 adolescent patients provided 220 continuous glucose-monitored (CGM) meals data from the two consecutive 3-day nutritional plans. Overall, the 50% carbohydrate diet was associated with higher postprandial glucose variability in the small meals (afternoon snacks, second breakfast) and greater postprandial peaks for other meals (breakfast, dinner, supper). Nevertheless, after the adjustment of a patient’s individual clinical variables (age, Tanner classification, glucose disposal rate), we observed that mean postprandial glucose was higher for afternoon snacks and lower for suppers in the 30% carbohydrate diet. Although a 30% carbohydrate diet seems to offer better postprandial glycemia, it requires additional attention from the physician and patient when it comes to modifying daily carbohydrate intake. Increased fat/protein content and size of the main meal lead to a prolonged postprandial glycemic response, which may affect the insulin treatment and result in suboptimal glycemic control.

## 1. Introduction

Type 1 diabetes mellitus is characterized by the autoimmune destruction of pancreatic beta cells, resulting in insufficient insulin production and the development of hyperglycemia [1]. Therapy for type 1 diabetes (T1DM) focuses on maintaining optimal blood glucose levels, achieved with intensive insulin treatment, proper nutrition, and physical activity. According to the principles of proper nutrition from Diabetes Poland, carbohydrates should comprise 45–60% of the daily energy requirement and the simple sugars contained in it should not exceed 10%. In a properly balanced diet, protein should comprise 15–20% of the total daily caloric intake, and fats should comprise 25–40% [2].

The standard of clinical care is to utilize self-monitoring of blood glucose and estimate the insulin requirement for the given meal [3]. However, achieving optimal therapy is challenging due to the complexity of matching insulin requirements for different meals and their quantity of carbohydrates, fats, and proteins. Nowadays, paying attention to insulinemic response along with glycemic response using an insulin index is much more effective in better metabolic control of diabetes [4,5]. In addition, other issues, i.e., patients’ fear of hypoglycemic episodes, affect optimal decision-making [6]. Those factors result in the increased rate of high blood glucose events such as postprandial hyperglycemia. In turn, suboptimal glycemic-control results in long-term diabetes complications, including nephropathy, retinopathy, neuropathy, and cardiovascular disease.

The dietary guidelines recommend the consumption of foods that provide a moderate and sustained postprandial glycemic response (PPGR) in order to minimize glucose excursions [7]. Recent scientific reports underline the importance of the composition of fats and protein, which, together with the glycemic index and glycemic load, play a pivotal role in the PPGR [4].

Although efficient and cost-effective, this method is still frequently associated with suboptimal insulin intake. Recent advancements, including the development of continuous glucose monitoring (CGM) and closed-loop systems, provided more in-depth information on the dynamics of postprandial glycaemia.

Herein, we wanted to investigate changes in postprandial glycemic responses (PPGR) in low (30%) and standard (50%) carbohydrate diets in adolescents with T1DM, taking into account both nutritional factors and the interindividual variability of patients.

## 2. Materials and Methods

Pediatric patients (aged 8–16) with at least a 1-year duration of T1DM and an HbA1c level ≤9.0% (75 mmol/mol) participated in a nutritional interventional study conducted in a Diabetes Reference Center in Gdansk, Poland. Patients with concomitant chronic diseases associated with special dietary requirements were excluded. Participants underwent CGM-monitored crossover nutritional intervention.

Every patient was introduced to two consecutive 3-day nutritional plans, with carbohydrates comprising 30% and 50% of the daily energy requirement. Before the intervention, each patient underwent a nutritional interview. On its basis, both diets were created following the nutritional standards for Polish children in 2021, taking into account the patient’s gender, age, and physical activity. Two consecutive 3-day nutritional plans were created with carbohydrates comprising 30% or 50% of the daily energy requirement. In planning the diets, attention was paid to the principles of a diet with a low glycemic index, including resistant starch or preparation of meals. Diets were composed and evaluated using the diet program Aliant and scored using the menu score.

The calories for both diets were calculated using Harris Benedict’s basal metabolic rate (BMR) formula for males = 66.47 + (13.7 × ideal body weight in kg) + (5 × height in cm) – (6.76 × age in years) and for females = 655.1 + (9567 × ideal body weight in kg) + (1.85 × height in cm) − (4.68 × age in years), respectively.

The Low Physical Activity Index was then added to the Total Metabolic Rate (TMR). On a carbohydrate-restricted diet, fats make up 40% and protein makes up 30% of the daily energy requirement, whereas on a 50% diet, fats make up 30% and protein makes up 20%, respectively. In the final phase, the daily energy requirement was divided into five meals, with the third meal (“dinner”) having the highest caloric value.

The nutritional plans were mainly composed of whole grain cereal products, e.g., rye bread, groats, and brown rice which accounted for 90% of the energy from carbohydrates, whereas simple sugars constituted up to 10% and came from fruit or natural dairy products. Raw or cooked vegetables appeared in almost every meal, which accounted for about 10% of the energy due to their low caloric value. A total of 40% of the energy from protein consisted of natural dairy products, e.g., yogurts and cottage cheese, and the rest was comprised of meat or fish. The fats in the diets were mainly of plant origin, e.g., nuts and vegetable oils. The carbohydrate content in the diets was kept at 3–5 g per kg body weight per day. The average fiber content in the individual diets was 30 g per day. The planned meals insulin index was less or equal to 29.

Both diets included five meals per day, served in 3-hour intervals from 7 AM to 7 PM. The second breakfasts were mostly whole grains with a portion of dairy, whereas the afternoon snacks consisted of low GI fruits with nuts or plain yogurts. All meals are detailed in Table 1.

Each patient participated in both nutritional interventions, one after another (cross-over and cross-sectional design), with no wash-out period in between. We implemented the 30%-carbohydrate diet in the hospital in order to avoid possible problems with maintaining normal glycemia with a low carbohydrate supply. Patients undergoing a 3-day diet plan with a carbohydrate content of 50% were allowed to follow this diet at home while under continuous supervision by the Principal Investigator using telemedicine solutions. For the 50% carbohydrate diet to be conducted at home, the parents were trained and informed about the diet plan and meal preparation, and remained in continuous contact with the Principal Investigator. Participants were ordered to avoid excessive physical activity during both the hospital stay and at home during the dietary intervention.

All patients were treated with a continuous subcutaneous insulin infusion therapy using Medtronic insulin pumps: Paradigm Veo 754 or 640G connected with Enlite CGM-RT and Guardian Sensor. Boluses and base insulin modifications were determined and administered by the diabetologist. Each patient was monitored for adverse events during the intervention and for 48 h before and after. After nutritional interventions, we manually paired each patient’s food intake, insulin therapy, and CGM data to evaluate postprandial glycemia under both diets. We defined meal CGM data as one obtained 30 min prior and 120 min after the meal. Each patient could theoretically provide data from 30 meals. We decided to include only 100% complete records and to exclude CGM data if additional insulin corrections were required for the meal. Further filtering was required to evaluate the inter-patient variability. To this point, we included only those patients who had paired meal CGM data in both diets for the given meal. For instance, if the patient had two breakfasts from the 30% carbohydrate diet and only one breakfast from the 50% carbohydrate diet, all three meals were included in the analysis. However, if the patient had records for three breakfasts from the 30% carbohydrate diet and none from the 50% one, the patient did not meet the above-mentioned criteria, and their data regarding breakfast meals were not analyzed further.

All patients underwent body composition analysis using a TANITA SC-240 MA. Standard laboratory tests were collected from each participant, including HbA1c, vitamin D, lipid profile, and liver enzymes. The glucose disposal rate was estimated using neural network approximation as described previously [8,9].

The statistical analysis encompassed the identification of critical postprandial glycemic variability metrics through principal component analysis and detrended fluctuation analysis. The selected parameters were compared between the diets using the univariate analysis and a generalized linear mixed model (GLMM) for the adjustment of patient-specific factors. A *p*-level of 0.05 was chosen as the threshold for significance.

## 3. Results

Due to the abovementioned filtering criteria, only data from 26 participants were available for analysis (Figure 1a). This group included 14 boys (53.8%) with a median age of 16 (11–17) years and 12 girls (46.2%) with a median age of 15 (14–16,5) years, respectively. The glucose disposal rate of patients was 6.08 (5.25–7.29) mg/(kg × min), BMI at 80.96th (57.95–89.94) centile and HbA1c of 7.2 (6.8–7.6) % (84 (74–89) mmol/mol), respectively (Appendix A). Patient glycemic control (Time-in-Range) did not significantly differ under the 30% vs. 50% carbohydrate diets (*p*-value 0.280; Appendix A) [7].

### 3.1. Analysis of All Complete Meal Data (n = 220)

First, using data from all meals with complete CGM recordings (*n* = 220, Figure 1a, Appendix A), we determined three main characteristics of postprandial glycemia: mean; coefficient of variation (CV%); and peak magnitude. We used those factors for downstream comparisons between diets. Analysis of meal data (*n* = 220, Table 1 and Appendix A) showed that the 50% carbohydrate diet resulted in a significantly higher overall postprandial CV% and glucose peak (*p* < 0.001, Figure 1b). Breakfasts, second breakfasts, and afternoon snacks had higher CV%, while breakfasts, dinners, and suppers had higher postprandial peaks in the 50% carbohydrate diet (Figure 1b; Appendix A).

### 3.2. Analysis of Meal Data in Patient-Paired Setup (n = 128)

Using paired meal data (Figure 1a; *n* = 128, Appendix A), we adjusted the data for patient-specific effects using GLMM. The patient-specific effects could be sufficiently modeled using the Tanner stage, age, lipid profile, and insulin sensitivity (R2 decreased from 0.75 to 0.71, *p* = 0.122). After adjusting for those patient-specific factors, the initially observed differences lost their significance. During the 30% diet, the afternoon snacks resulted in a higher mean postprandial glucose, while suppers presented a significantly higher mean postprandial glucose during the 50% diet. Using GLMM, we also determined that the starting blood glucose and meal size were significant modifiers of mean postprandial glycemia independent of patient-specific factors (R = 0.54 and 0.10, *p* < 0.001).

## 4. Discussion

The macronutrient content may significantly influence postprandial glycemia. Successful modification of carbohydrate intake requires careful investigation of CGM, a diet plan, and patient clinical data. Despite a homogeneous study population, we observed that differences across patients influenced the interpretation of postprandial glycemia. Fortunately, those differences can be reliably modeled using factors associated with insulin response.

Despite the content of the meal, the metabolic and hormonal response after a meal is also crucial for diabetes management. The degree of insulin sensitivity is one of the main factors in the control of postprandial glycemic response dynamics [7]. Interestingly, a decreased insulin response or insulin resistance is often observed in adolescents with whom growth and sex hormones play a decisive role. It is well known that the former has an anti-insulin effect, as it weakens the ability of insulin to stimulate glucose uptake in peripheral tissues and enhances hepatic glucose synthesis [10,11]. Therefore, we recommend evaluating the patient’s Tanner stage, lipid profile, and glucose disposal rate for optimal glycemic control during the dietary intervention.

Continuous glucose monitoring is an excellent tool providing additional information of patients’ glycemic control by indication of time spent in the target glucose range. Together with HbA1c, they accurately reflect glycemic control. The dietary intervention (30% vs. 50% carbohydrate diet) had no significant effect on the glycemic control observed in patients prior to the study. This may be due to the short duration of the study, and longer prospective studies are required to fully detect any clinically relevant change.

A significant limitation of our study was using only a simple bolus, which may have limited utility in a 30% carbohydrate diet due to increased fats and protein content. Despite optimal insulin therapy, increased meal size and fat–protein contents were a modifier of postprandial glycemic response (PPGR). In fact, high-fat, protein-rich meals may result in a delayed postprandial glycemic elevation by 3–12 h [12]. This bears implications for clinical practice, as patients are often advised to self-monitor their blood glucose levels ~2 h after the meal [4]. We suspect that the observed (after patient-specific adjustment) elevation of mean postprandial glucose for afternoon snacks in the 30% carbohydrate diet could result from the higher fat and protein content in the dinner meals. Moreover, an elevated starting blood glucose at supper in a 30% carbohydrate diet could lead to insulin bolus over-correction, resulting in overly lowered mean postprandial glucose after supper. Similar studies suggest increasing the insulin dose for high-fat meals by up to 20% and application of a dual-wave dose to optimize the glycemic response [4]. Although there are methods for calculating an additional dose for high protein/fat meals, they do not appear to be practical for daily use and carry an increased risk of hypoglycemia [12].

Future studies of the 30% carbohydrate diet should focus on the level and time after each meal of the postprandial glucose peaks to prevent a hyperglycemic effect. In addition, an interindividual insulin bolus algorithm covering high protein/fat content to optimize therapy approach for T1DM adolescents should also be considered.

## 5. Conclusions

While a 30% carbohydrate diet presents overall better postprandial glucose, due to its higher fat/protein meal content it requires additional attention from the physician and the patient. A prolonged postprandial glycemic response, often observed after main meals, may lead to overlaps and impede optimal glycemic control.

## Figures and Tables

**Figure 1 nutrients-14-04689-f001:**
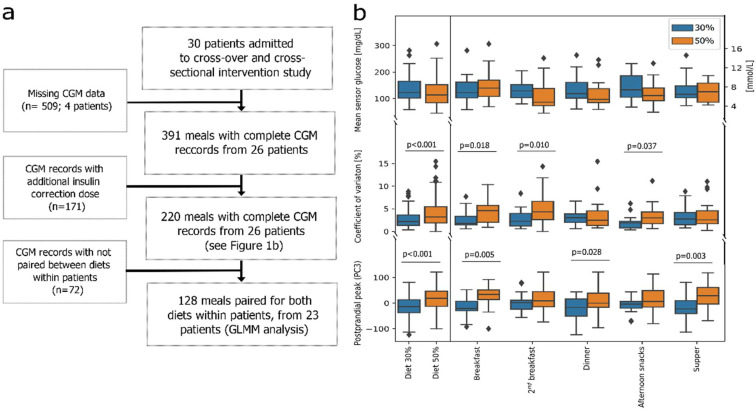
(**a**) Study flowchart; (**b**) Comparison of mean sensor glucose, coefficient of variation and postprandial peak between diets and meals for 220 observations. CGM—continuous glucose monitoring; GLMM—generalized linear mixed models; PC3—principal component 3 (see Appendix A). Outliers (>3SD) were marked with ◆.

**Table 1 nutrients-14-04689-t001:** Meal and insulin-therapy related factors for meals analysis (*n* = 220 records). Values are provided as % and number of cases or mean ± standard deviation.

Meal	Diet	Caloric Value (kcal)	Meal Weight (g)	Hydrated Carbon (CHO) (%g)	Carbohydrate Exchange Unit	Fat-Protein Exchange Unit	Fats (g)	Proteins (g)	Meal Insulin Requirements (U)
Breakfast (7:00 AM)	30% *n* = 20	432.50 ± 59.66	290.50 ± 51.65	37.00 ± 6.16	3.68 ± 0.59	3.55 ± 0.25	19.50 ± 2.24	44.75 ± 1.12	5.29 ± 2.73
	50% *n* = 24	409.17 ± 46.92	263.33 ± 44.59	49.17 ± 7.17	4.98 ± 0.73	2.17 ± 0.38	11.67 ± 3.81	15.42 ± 11.41	7.92 ± 3.02
Second Breakfast (10:00 AM)	30% *n* = 21	332.86 ± 140.84	173.33 ± 120.51	25.95 ± 12.41	2.67 ± 1.21	1.91 ± 1.39	5.48 ± 1.50	10.48 ± 1.50	3.44 ± 3.09
	50% *n* = 27	267.22 ± 34.57	203.70 ± 25.29	25.37 ± 6.34	2.31 ± 0.54	0.90 ± 0.10	5.00 ± 0.00	27.04 ± 2.50	3.06 ± 1.50
Dinner (1:00 PM)	30% *n* = 20	446.00 ± 66.24	318.75 ± 92.05	38.00 ± 12.81	3.83 ± 1.23	3.33 ± 1.38	27.50 ± 2.56	10.00 ± 0.00	4.54 ± 2.08
	50% *n* = 24	523.75 ± 71.86	447.50 ± 27.54	68.96 ± 8.07	7.00 ± 0.85	3.00 ± 0.40	20.63 ± 1.69	10.00 ± 0.00	8.90 ± 4.33
Afternoon snack (4:00 PM)	30% *n* = 20	219.50 ± 81.21	182.75 ± 28.54	23.00 ± 7.33	2.33 ± 0.69	1.22 ± 0.44	5.25 ± 1.12	45.00 ± 0.00	2.67 ± 1.39
	50% *n* = 19	171.58 ± 21.15	228.95 ± 20.25	23.16 ± 5.82	2.21 ± 0.48	0.86 ± 0.10	5.00 ± 0.00	26.84 ± 2.99	2.62 ± 1.69
Supper (7:00 PM)	30% *n* = 25	326.80 ± 61.68	165.60 ± 47.62	30.60 ± 8.08	3.08 ± 0.77	1.37 ± 0.70	10.40 ± 7.35	9.80 ± 2.27	3.94 ± 2.14
	50% *n* = 20	346.00 ± 41.09	214.00 ± 19.57	47.00 ± 9.79	4.25 ± 0.47	1.66 ± 0.42	11.75 ± 2.45	10.00 ± 0.00	4.81 ± 2.79

## Data Availability

Data presented in this study are available on request from the corresponding author. The data are not publicly available due to subjects’ privacy.

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
