# Peer review of "Reduced Carbohydrate Diet Influence on Postprandial Glycemia—Results of a Short, CGM-Based, Interventional Study in Adolescents with Type 1 Diabetes"

_nutrients, 2022, doi:10.3390/nu14214689_

Round 1

Reviewer 1 Report

This brief report presents an investigation of postprandial blood glucose changes with low (30%) and standard (50%) carbohydrate diets in adolescents with T1DM. Since the treatment of type 1 diabetes (T1DM1) focuses on maintaining optimal blood glucose levels through intensive insulin therapy, adequate nutrition and physical activity, it is of interest to analyse in detail how diet composition may influence postprandial blood glucose. Therefore, this study is considered of interest for publication.

Furthermore, it adequately details the methodology followed both in the design of the nutritional intervention and in the data collection and subsequent statistical treatment of the data. For this reason, the results are adequately presented as well as their corresponding discussion and conclusion.

The bibliography is also adequate given that it is a short report.

Perhaps, the summary presentation of the study makes the introduction as well as the discussion and conclusions too brief and I would encourage the authors to present a more detailed study with a larger number of participants for a future publication.

Author Response

Thank you for your suggestion. We added more crucial information to the introduction as well as the discussion section to better illustrate the aim of our study. 

Reviewer 2 Report

Q1: Abstract section was suggested to be further improved, to make a better summary of the full text.

Q2: How are the 30% and 50% carbohydrate paired-meals analysis represented in Table 1? What is the design basis? Are there detailed calculation instructions of 30% and 50%?

Q3: 3.1 section, what’s the meaning of coefficient of variation (CV%)? What did these values mean, and what hints do authors have about the results of this article, please supplement relevant contents.

Q4: In this report, the values of N fluctuated greatly in these results. Why can't authors select the information of several patients with complete data for analysis? I assume that the changes in the number of patients would affect the rigor of the full-text results.

Q5: In order to demonstrate the effective research value of this report, the discussion part expressions are not sufficient and need to be further improved.

Q6: How to reflect the research significance of this report? The effects of 30% and 50% carbohydrate intake on blood glucose in diabetic patients are predictable, please improve the relevant statements.

Author Response

Q1: Abstract section was suggested to be further improved, to make a better summary of the full text.

A1: We revised and expanded the abstract to better explain the aim of our study.

Q2: How are the 30% and 50% carbohydrate paired-meals analysis represented in Table 1? What is the design basis? Are there detailed calculation instructions of 30% and 50%?

A2: In order to keep clarity and scheme we followed during the statistical analysis (see flowchart), firstly we presented meal analysis for all 220 observations. The paired meal analysis was included in the supplement. The process and formulas of calculating the 30% and 50% diets were added to the main text.

Q3: 3.1 section, what’s the meaning of coefficient of variation (CV%)? What did these values mean, and what hints do authors have about the results of this article, please supplement relevant contents.

A3: The coefficient of variation (CV%) is a fairly standard Continuous Glucose Monitoring metric, and equals to standard deviation of sensor glucose values divided by mean sensor glucose. As standard deviation is typically correlated with the mean, the purpose of the CV is to standardize the variability and make it comparable between patients with different average glucose levels. Thus, CV values reflect the dispersion of sensor glucose value – the higher the CV%, the less stable the glucose values, and the more possible it is for patients to experience extremely high and low sensor glucose values. This parameter is discussed in depth in Battelino et al. 2019 [ref. 9 in the main text].

Q4: In this report, the values of N fluctuated greatly in these results. Why can't authors select the information of several patients with complete data for analysis? I assume that the changes in the number of patients would affect the rigor of the full-text results.

A4: This issue was troublesome, and we revised several sections of the manuscript to better explain and justify our approach. The analysis could be performed purely on a meal-by-meal basis – which resulted in a high number of data points and, through a high power of such comparisons allowed us to quantify the impact of different diets. However, the intrinsic (genetic, clinical, social etc.) factors inherent to each of the patients could be perceived as a potential confounding factor. Having the meal-by-meal data confounded by such factors could skew the results, and for this reason, we considered an analysis adjusting for the individual-specific factors. This however modified the number of available data points due to the statistical constraints of a mixed-effects model. Having both these aspects was important to encompass the whole scope of meal-related GV parameters and is relevant for potential future studies, indicating how patient adherence and sensor technology result in a decreased number of observations given a specific design. Section 3.1 reflect most studies, and indicates that some relevant differences (observed in section 3.2 - paired design) may be missed.

Q5: In order to demonstrate the effective research value of this report, the discussion part expressions are not sufficient and need to be further improved.

A5: We expanded the scope of discussion to underline the clinical relevance and perspectives of our report.

Q6: How to reflect the research significance of this report? The effects of 30% and 50% carbohydrate intake on blood glucose in diabetic patients are predictable, please improve the relevant statements.

A6: Our report investigates postprandial blood glucose changes with low (30%) and standard (50%) carbohydrate diets in adolescents with T1DM. Although a 30% carbohydrate diet seems to have an overall better postprandial glucose, we observed that due to its high protein/fat component, it leads to the prolonged postprandial glycemic response, resulting in an overlapping effect after the afternoon snacks. Therefore, due to the elevated starting blood glucose at supper, we tend to give the patients over-correction bolus, which in turn causes overly lowered mean postprandial glucose after the last meal of the day. This observation is of substantial clinical importance as its changes the perspective of the 30% carbohydrate diet and its impact on glycemic control.

Round 2

Reviewer 2 Report

Thank you for these modifications, I have no any other concerns.

Author Response

We appreciate the positive comments from the Reviewer. We corrected the manuscript to improve the English language, as requested.